# Assigning Defined Daily/Course Doses for Antimicrobials in Turkeys to Enable a Cross-Country Quantification and Comparison of Antimicrobial Use

**DOI:** 10.3390/antibiotics10080971

**Published:** 2021-08-12

**Authors:** Philip Joosten, Steven Sarrazin, Claire Chauvin, Gabriel Moyano, Katharina Wadepohl, Liese Van Gompel, Jaap A. Wagenaar, Jeroen Dewulf

**Affiliations:** 1Department of Obstetrics, Reproduction and Herd Health, Faculty of Veterinary Medicine, Ghent University, 9820 Merelbeke, Belgium; s.sarrazin@lammerant.be (S.S.); jeroen.dewulf@ugent.be (J.D.); 2Anses, French Agency for Food, Environmental and Occupational Health and Safety, 22440 Ploufragan, France; Claire.CHAUVIN@anses.fr; 3Departamento de Sanidad Animal y Centro de Vigilancia Sanitaria Veterinaria (VISAVET), Facultad de Veterinaria, Universidad Complutense de Madrid, 28040 Madrid, Spain; gmoyano@ucm.es; 4Institute of Food Quality and Safety, University of Veterinary Medicine Hannover, 49456 Bakum, Germany; katharina.wadepohl@tiho-hannover.de; 5Institute for Risk Assessment Sciences, Utrecht University, 3584 CM Utrecht, The Netherlands; L.VanGompel@uu.nl; 6Department of Bacteriology and Epidemiology, Wageningen Bioveterinary Research, 8221 RA Lelystad, The Netherlands; J.Wagenaar@uu.nl; 7Department of Infectious Diseases and Immunology, Faculty of Veterinary Medicine, Utrecht University, 3584 CL Utrecht, The Netherlands

**Keywords:** antimicrobial use, antimicrobial resistance, turkeys, antimicrobial quantification, treatment incidence

## Abstract

Antimicrobial resistance (AMR) threatens our public health and is mainly driven by antimicrobial usage (AMU). For this reason the World Health Organization calls for detailed monitoring of AMU over all animal sectors involved. Therefore, we aimed to quantify AMU on turkey farms. First, turkey-specific Defined Daily Dose (DDD_turkey_) was determined. These were compared to the broiler alternative from the European Surveillance of Veterinary Antimicrobial Consumption (DDD_vet_), that mention DDD_vet_ as a proxy for other poultry species. DDD_turkey_ ranged from being 81.5% smaller to 48.5% larger compared to its DDD_vet_ alternative for broilers. Second, antimicrobial treatments were registered on 60 turkey farms divided over France, Germany and Spain between 2014 and 2016 (20 flocks per country). Afterwards, AMU was quantified using treatment incidence (TI) per 100 days. TI expresses the percentage of the rearing period that the turkeys were treated with a standard dose of antimicrobials. Minimum, median and maximum TI at flock level and based on DDD_turkey_ = 0.0, 10.0 and 65.7, respectively. Yet, a huge variation in amounts of antimicrobials used at flock level was observed, both within and between countries. Seven farms (12%) did not use any antimicrobials. Aminopenicillins, polymyxins, and fluoroquinolones were responsible for 72.2% of total AMU. The proportion of treating farms peaked on week five of the production cycle (41.7%), and 79.4% of the total AMU was administered in the first half of production. To conclude, not all DDD_vet_ values for broilers can be applied to turkeys. Additionally, the results of AMU show potential for reducing and improving AMU on turkey farms, especially concerning the usage of critically important antimicrobials.

## 1. Introduction

Many components of our human and animal healthcare depend on the availability of effective antimicrobials. Thanks to these antimicrobial drugs, bacterial infections have become far less life threatening [1]. However, the extensive and often unnecessary use of antimicrobials has driven the selection for antimicrobial resistance (AMR) [2]. Nowadays, resistant infections are responsible for 700,000 deaths in the world, annually [3]. AMR also impacts the field of veterinary medicine, as illustrated by therapy failures often observed in animals [4] or the transmission of resistant bacteria between farm animals and farmers [5]. This has led to the recognition by the World Health Organization (WHO) of AMR as one of the biggest public health threats globally [1]. To address this issue and to ensure that antimicrobials stay effective when they are needed, the total antimicrobial consumption should be reduced. Several studies have already proven the direct link between antimicrobial usage (AMU) and AMR [6]. As a result, many national and international initiatives have been set up in the past decade to monitor and reduce AMU with the aim of reducing the prevalence of AMR [7,8].

In Veterinary Medicine, the first international report on AMU in Europe was published in 2010 by the European Surveillance for Veterinary Antimicrobial Consumption (ESVAC), which is part of the European Medicines Agency (EMA). Back then, this report included sales data in tonnes of active substance (AS) from 9 European countries for the period between 2005 and 2009 [9]. The latest report includes the same type of data but covering 31 countries [8]. The ESVAC reports have shown that the sales of antibiotics for use in animals has declined more than 34% between 2011 and 2018 [8]. However, the national sales data presented in the ESVAC reports have limitations. Reporting AMU as the total weight of AS sold does not take into account dosing differences between products, nor does it allow to stratify AMU by species or weight group (e.g., weaned piglets vs. finisher pigs). In addition, the ESVAC’s indicator includes all biomass produced at country level in the denominator, which is highly influenced by the composition of the national animal population [10]. These limitations are avoided when collecting AMU at farm and species level. Moreover, collecting reliable AMU data at the level of the end-user is essential for the transition to a lower and more responsible AMU in food animal production, as proven by the systems set up in several countries [7]. For this reason, ESVAC published a paper in 2013 to guide countries in setting up monitoring systems at farm level and by animal species [11]. Over the past years, many countries have launched such systems, with 38 active farm-level AMU monitoring systems in 16 countries in March 2020 [7].

However, minor species in food animal production, such as turkeys, are often overlooked when setting up AMU monitoring systems. AMU at turkey farms has so far only been described in Canada (2013) [12], the Netherlands (2013) [13], France (2004, 2018) [14,15], Italy (2019) [16] and Austria (2013–2019) [17]. In addition, differences between systems regarding data collection, sampled farms, quantification of AMU, etc. often prevent supranational comparison. To promote harmonization of AMU quantification, ESVAC published Defined Daily Doses (DDD_vet_) and Defined Course Doses (DCD_vet_) for animals. Yet again, the focus was, understandably, put on the major food production animals with DDD_vet_ and DCD_vet_ values for pigs, cattle and broilers. ESVAC does mention that the latter can be used as a proxy for other poultry species [18]. However, to the best of our knowledge, multi-country studies at farm level, using the broiler DDD_vet_ or DCD_vet_ values to quantify AMU in turkeys, are non-existent. At the same time, these are necessary, knowing that 14.8% of poultry meat in Europe comes from the turkey industry [19] and European legislation will make monitoring of AMU at farm level mandatory by 2030 for all food animal species, including turkeys [20].

Therefore, the aim of this study was to determine defined daily doses (DDD_turkey_) and defined course doses (DCD_turkey_) specifically for turkeys based on all antimicrobial products registered for use in turkeys in France, Germany and Spain. Based on these values, AMU was quantified in a standardized manner at farm level in 60 conventional turkey farms in France, Germany and Spain (20 farms/country).

## 2. Results

### 2.1. Assigning Defined Daily Doses Turkey (DDD_turkey_) and Defined Course Dose Turkey (DCD_turkey_) for France, Germany and Spain

#### 2.1.1. DDD_Turkey_ and DCD_Turkey_

In total, there were 255 products licensed for use in turkeys, of which 128 in France, 61 in Germany, and 66 in Spain. Eighty-eight percent (*n* = 224) of the products were registered for use in water. Ten percent (*n* = 25) and two percent (*n* = 6) of the products were authorized for use in feed or parenteral use, respectively. In total, 91% of the licensed products in this study are listed as critically important antimicrobial (CIA) for veterinary use. Fifty-five percent are listed as CIA for human medicine (see Appendix A). Based on the AS and administration route of the products, a total of 28 unique categories were identified, for which a DDD_turkey_ and a DCD_turkey_ were calculated (Table 1). There were 19 different active substances, including combinations of ASs (*n* = 4). For all 19 ASs, there were products registered for use in water. Four and five ASs also had products registered for parenteral use or for use in feed, respectively. This includes colistin and oxytetracyline with products registered for all three administration routes.

When ASs authorized products for both oral administration and injection, the mean dosages for the parenteral administration were lower than the ones for the oral administration having the same AS (Table 1). Doxycycline, colistin and enrofloxacin had the most authorized products with 37, 36 and 29 authorizations, respectively. DDD_turkey_ ranged from being 81.5% smaller than its DDD_vet_ alternative to 48.5% larger than the corresponding DDD_vet_ value. Although there were great differences between corresponding DDD_turkey_ and DDD_vet_ values, they also showed similarity. For example, DDD_turkey_ for enrofloxacin and tylvalosin products authorized for use in water did not differ from their corresponding DDD_vet_ value for broilers. On average DDD_turkey_ values were 8.8% lower compared to their corresponding DDD_vet_ for broilers. A complete overview is visualized in Figure 1. However, benzylpenicillin and phenoxymethylpenicillin are not included, as the former has no corresponding DDD_vet_ value and the latter did not have products registered for use in turkeys.

#### 2.1.2. Deviation from DDD_Turkey_

Appendix A gives an overview of the deviations from the mean consensus DDD_turkey_ within each of the 28 categories stratified by AS and administration route. Within 15 of the 28 categories (54%), none of the products showed a deviation of 10% or higher from the consensus DDD_turkey_. The top 20 deviating products are shown in Table 2. Half were registered in Spain. Thirteen were products with oxytetracyline (*n* = 9) or doxycycline (*n* = 4) and 6 products were authorized for use in feed.

##### Parenteral Products

All parenteral products were French (Appendix A) and are on the list of CIAs for veterinary use from the World Organisation of Animal Health (OIE). Within the four ASs, where the active substance is authorized for parenteral administration, two (ampicillin and oxytetracycline) were represented by one product, and two (colistin and a combination of lincomycin and spectinomycin) showed harmonization in recommended dose.

##### Feed Products

Of the 25 products registered for use in feed, eighteen, six and one were registered in France, Germany and Spain, respectively (Appendix A). All products can be grouped into five unique categories based on their AS. Three categories (sulfadiazine+trimethoprim (*n* = 3), tiamuline (*n* = 4) and tylosin (*n* = 6)), did not contain registered products that deviated >10% from the consensus DDD_turkey_. Within these categories, the registered products also showed (almost) perfectly harmonized recommended doses (Appendix A). For colistin products authorized for use in feed (*n* = 3), 33% of the products deviated > 10%. All six of the Spanish products were products with oxytetracycline, which together had a mean recommended dose of 7.4 mg/kg of animal weight. The other three products, within the category of oxytetracycline authorized for use in feed, were French and had a mean recommended dose of 40 mg/kg of animal weight. The difference between the lowest and highest recommended dose within this category equaled 784% (Appendix A). As a result, all of the oxytetracyline products (*n* = 9) deviated >10% from the mean consensus value of that category.

##### Water Products

Fourty-two percent of the categories for which the active substance was authorized for use in water (*n* = 19) did not contain products that deviated >10% from the mean consensus value of that category. In two of these 19 categories, the consensus DDD_turkey_ was based on just one product. Nevertheless, within the category of enrofloxacin products authorized for administration via water, the recommended doses were perfectly harmonized. Contrastingly, tiamulin products authorized for use in water (*n* = 12) did not show consistency in recommended dose within and between countries. As a result, all products in this category deviated >10% from the consensus DDD_turkey_. Similar results were found for five other categories: colistin, flumequin, lincomycin+spectinomycin, neomycin and oxytetracycline (Appendix A).

### 2.2. Quantification of Antimicrobial Usage at Flock Level on 60 Turkey Farms in France, Germany and Spain

The 20 farms in each country were not randomly sampled and cannot be considered representative for the turkey production in a country (an overview of production parameters of the sampled flocks can be found in Appendix A, including an overview in Appendix A-bis on the size of all poultry holdings other than broilers and laying hens in country B, E and H). For this reason, countries were anonymized by assigning them a letter, as agreed by the different partners within the Ecology from Farm to Fork Of microbial drug Resistance and Transmission (EFFORT) consortium (letters B, E and H). The authors would like to emphasize that there are differences between the licensed products of a country and the used products in that country. So, when certain products are only licensed in one country, usage of these products did not necessarily take place in that country.

#### 2.2.1. AMU Quantification at Flock Level

Overall, 53 farms administered 230 antimicrobial group treatments. At country level there were a total of 67, 69 and 94 treatments in country B, country E and country H, respectively. Seven farms, of which three in country E, three in country B and 1 in country H, did not use any antimicrobial group treatments in the flock of turkeys from which the data was collected. The median TIDDD_turkey_ at flock level (TIDDD_turkey_-FL) equaled 10.0. and varied from 0.0 to 65.7. Meaning that the median farm in this study was treating the flock with a standardized dose of antimicrobials during 10% of its rearing period. TIDDD_turkey_-FL of the two flocks from country H with the highest usage equaled 55.5 and 65.7 and was more than twice as high as the maximum TIDDD_turkey_-FL in country B and country E (Table 3 and Appendix A). An overview of the variation in AMU at flock level, both within and between countries is shown in Table 3 and Figure 2.

#### 2.2.2. Qualitative Description of AMU at Flock Level

##### Administration Route

From the 230 treatments, 210 were administered through water, 12 were administered through feed and 8 were parenteral administrations. Farmers from country B and country E only administered treatments through water. All feed and parenteral administrations were treatments on farms from the same country. However, the portfolio of antimicrobial products with a marketing authorization for use in turkeys within a country was not always mirrored by the used products in that country. The parenteral treatments included 4 treatments with colistin and 4 treatments with the combination of lincomycin and spectinomycin. Together they represented 3.7% of all usage registered on the flocks from country H. The treatments through feed were 5 tylosin treatments and 7 oxytetracycline treatments, representing 7.5% of the total registered AMU on the farms from country H. In relation to the total AMU reported in this study, 5.4% is represented by parenteral treatments (1.9%) and treatments through feed (3.6%).

##### Antimicrobial Classes

Together, the extended-spectrum penicillins (ES penicillins), polymyxins and fluoroquinolones represented 72.2% of all AMU in this study. No cephalosporins were used within the sampled flocks. The ES penicillins were represented by 52 amoxicillin treatments and 20 ampicillin treatments. The polymyxins in this study were all treatments with colistin (*n* = 35), while the fluoroquinolone treatments were made up by 28 enrofloxacin and one flumequin treatments. On country level the ES penicillins, polymyxins and fluoroquinolones did not always belong to the three largest classes. They represented 43.9% of all AMU on the farms from country B, compared to 69.9% and 86.0% for country E and country H, respectively. In country B, tetracyclines were the largest class, with 18.3% of the total AMU, whilst β-lactamase sensitive penicillins (βLS penicillins) were the most frequently used class, with 23 treatments (22 benzylpenicillin treatments and 1 phenoxymethylpenicillin treatment). There was no usage reported of the latter class on the farms of country E or country H. On the farms of country E, macrolides were the third largest class used, after ES penicillins and polymyxins. Moreover, fluoroquinolones were only used three times and represented 5.0% of the total AMU of country E. In country H, almost half (47.8%) of the total AMU were treatments with amoxicillin (ES penicllins). This equaled 25.0% and 13.2% for the fluoroquinolones and polymyxins, respectively. A complete overview of the AMU distribution over the different classes in total and per country is presented in Figure 3 and Appendix A.

##### Indication for Treatment

Intestinal disorders (60.5%), respiratory disease (28.9%) and colibacillosis (7.9%) were the most common indications for treatment. Only six treatments had another indication. Firstly, three treatments were initiated to treat locomotive disorders on two farms in country E. Secondly, on two other farms in country E, non-specific disorders and higher mortality rate were indications reported by the farmer. Finally, one farm in country H set up a treatment for general disorders. For two of the 230 treatments, there was no indication for the treatment reported. An overview of the treatment indications stratified by country and in total is presented in Appendix A.

##### AMU over the Period of one Production Cycle

Five out of the 60 farms (8.3%) set up a treatment on the first day of production. In the entire first week, 11 farms had initiated at least one treatment. The proportion of farms treating the animals hit 30% in week 2, increased to a maximum of 41.7% in week 5 and plateaued around 30% from week 6 until week 9 to decrease to 3.3% from week 16 to week 19 (Figure 4). By the end of week 9, 178 treatments were initiated, representing 79.4% of all AMU (=sum of all TIDDD_turkey_) reported in this study. This higher usage in the first half of production was present in all countries, with 61.7%, 76.8% and 88.5% of all AMU being administered in the first 9 weeks in country B, country E and country H, respectively. Treatments initiated closest to the end of production were registered on two farms from country B at day 124 and day 126 of production (both in week 18). Both treatments lasted for 5 days and ended in the beginning of week 19. For the farms from country E and country H, the treatment day closest to the end of production, was registered on day 114 and day 102, respectively.

The indication for treatment varied with the age of the turkeys. Of the 138 intestinal treatments, 72.5% was initiated between week 2 and week 8. A respiratory indication was most common from week 4 to 12, covering 75.8% of the 66 reported respiratory treatments. Colibacillosis as indication for treatment peaked on week 5 with 29.4% of the 17 treatments. Seven other treatments for colibacillosis (41.1%) were reported from week 7 to week 11 (Appendix A). Not only the indications for treatment changed throughout the rearing period, this was also the case for the type of antimicrobial class that was used. ES penicillin usage peaked in the first half of production with 84.7% of its treatments reported in the first 9 weeks. Usage of other classes also peaked in the first half of production, but were not so frequently used in the first weeks. For example, 78.6% of all fluoroquinolone treatments were registered from week 5 to 10. This equaled 100% for the combination treatments of lincomycin and spectinomycin over the same period. βLS penicillin usage peaked later on in production, as 56.5% of its usage was registered between week 7 and week 13 (Appendix A).

## 3. Discussion

### 3.1. Establishing Defined Daily Doses

This multi-country study reports on DDD_turkey_ and DCD_turkey_ values based on all antimicrobial products authorized for use in turkeys in France, Germany or Spain. These values might serve as a more suitable alternative to quantify AMU on turkey farms in Europe than the DDD_vet_ and DCD_vet_ values from ESVAC, that are focused on broilers. However, ESVAC does state that these values can be used as a proxy for turkeys [18], although to date no other studies have used these values for AMU quantification on turkey farms, as far as the authors are aware. Let alone, these studies looked into the suitability of these values for application in turkeys. One Canadian [12] and one Italian [16] study however, did calculate their own DDD_vet_ values. However, both single-country studies used only one list of DDD_vet_ values for both broilers and turkeys combined. Preferably, DDD values are assigned by unique species and based on the antimicrobial products registered in multiple countries [10,21]. The latter is rather limited in this study, with only three countries. Including more countries, especially those with a large turkey production such as Italy and Poland, would be beneficial for assigning European DDD values to quantify AMU on turkey farms.

To establish a list of standardized doses, it always demands generalization and simplifications [10,21,22]. The big variation in recommended dose within a category of products, with the same combination of ASs and administration route, causes some products to deviate strongly from the standardized doses (Table 2). As a result, DDD values will not always reflect the actual used dose at flock level and should therefore not be used to discuss under- and overdosing. However, they do allow for the use of certain indicators to quantify AMU, that enable comparisons between flocks, farms and, most importantly, countries. Furthermore, using DDD-based indicators should be preferred when quantifying AMU, as these give correct insights into the relative importance of active substances and administration routes [10].

Other aspects of dosing, such as the agreement with the recommended dose or variation caused by differences in formulation between products with the same AS, are difficult to take into account when establishing DDDs. Improper dosing is an issue [23,24], but agreement with the recommended dose remains difficult to report [22]. Differences in formulation of the same AS can lead to variation between the recommended doses from the summary of product characteristics (SPC). However, to avoid additional complexity in listing DDDs, it was decided to only work at the level of active substance and not take into account different molecular formulations.

It has to be mentioned that the antimicrobial portfolio within a country is subject to changes, as older products are taken off the market, while new products are added. The DDD_turkey_ values reported in this paper, are based on the 255 products authorized for use in turkeys in France, Germany and Spain in July 2017. Changes to this portfolio of antimicrobial products might have occurred since then. However, as the number of authorized products was substantial for the majority of AS categories, updates to this portfolio will likely have little impact on DDD values of AS categories with a substantial number of products, as DDDs represent mean values.

Calculating DDDs as a mean, as suggested by ESVAC [21] and previous studies [22,25], does add weight to countries with a larger antimicrobial drug portfolio, as a result they will have a bigger impact on the DDD values. In this study, France had 128 registered products, compared to 61 and 66 in Germany and Spain, respectively. However, this trend was not applicable for all AS categories, as in 54% of them, none of the products showed a deviation of 10% or higher from the consensus DDD_turkey_. To allow for a good insight in this matter, Appendix A includes mean, median, mode and deviation from the consensus DDD_turkey_.

All registered antimicrobials were registered for administration via one of three administration routes: injection (parenteral), feed or water. DDD_turkey_ values in this study were calculated separately for products administered via water and feed, instead of analyzing them together as oral administrated products, as done by ESVAC for broilers [18] and Postma and colleagues [22]. However, both state that dose differences between water and feed administered products with the same AS can be huge for pigs. This is also the case in this study for products with sulfadiazine in combination with trimethoprim (Table 1). As a result, splitting feed and water administrations enables more precise DDDs and a more detailed AMU analysis.

### 3.2. Variation in Recommended Doses

Within certain AS categories, the recommended dose of the similar products showed large variation, both within and between countries. As a result, some products deviated strongly from the DDD value. Furthermore, this can lead to big differences between international DDDs and national DDDs but also among national DDDs [22,26]. The most extreme deviation was present for oxytetracyline products registered for use in water. As a result, three products of this category ended up in the top 20 of most deviating products (Table 2). Such extreme differences will influence AMU calculations, depending on what type of dose parameter is being used in the AMU indicator. However, when the goal is to report on AMU cross-country, international harmonized DDD values are indispensable [21,22].

As only half (54%) of the AS categories showed harmonization among the registered products, this study shows the need for European-wide harmonization when it comes to recommended doses of similar antimicrobial products, authorized for use in turkeys. In 2015, Postma and colleagues already made a similar conclusion for antimicrobial products registered for use in pigs [22]. EMA wants to address this and therefore published a report that describes a non-experimental methodology for dose review and adjustment of established veterinary products [27]. However, not all AS showed this variation. Due to a European legislation, article 34 of Directive 2001/82/EC [28], enrofloxacin products registered for use in water (*n* = 29) for instance, were perfectly harmonized, regardless of the high number of authorized products in all three countries. The latter shows that harmonization is possible, both within and between countries. Most likely, “country” and “number of years since authorization” do have an influence on the recommended dose as discussed by Postma et al. [22]. To select the products in need for a dose review, ESVAC proposes a list of criteria, one of them being the existence of different dose recommendations within a group of similar products [27]. The authors believe that studies like this one can help guide prioritization and selection of candidates.

### 3.3. Comparision of DDD_turkey_ and DDD_vet_

Differences in recommended dose are not only country-related, but also species-related in the case of similar products authorized for use in a different (poultry) species. DDD_turkey_ values from this study deviated between −81.5% and +48.5% from ESVAC’s DDD_vet_ for broilers (Figure 1). For example, “ampicillin C20” and “ampicillin P 1000mg/g” are both products registered for use via water in broilers in Germany, with a recommended dose of 160 and 320 mg/kg body weight, respectively (data not shown) [29]. In France, four ampicillin products registered for use via water in turkeys, all have a recommended dose of 20 mg/kg body weight (Appendix A) [30]. This big difference might partially be explained by non-species factors, such as year and country of authorization. However, as different studies point out that pharmacokinetic characteristics can be significantly different between turkeys and chickens [31,32], caution is needed when using broiler DDD values, such as ESVAC’s DDD_vet_, to quantify AMU on turkey farms. Nevertheless, the ESVAC values are based on all antimicrobial products registered for use in broilers in at least one of nine European Union Member States, compared to only three countries for the standardized doses reported in this study. As a result, the standardized doses from ESVAC will represent more products, which might better balance out international dosing differences between similar products. A comparison between DDD_turkey_ and a DDD_broiler_ value based on products from the same three countries, would simplify the comparison. In addition, many products registered for use in turkeys are often also registered for use in broilers. As a result, DDD_turkey_ values from enrofloxacin and tylvalosin did not deviate from their DDD_vet_ alternative for broilers. Therefore, the consequences for AMU quantification when using one or the other dose parameter should be further examined. Most likely it will be dependent on the relative importance of certain ASs that show a big deviation between DDD_turkey_ and DDD_vet_.

### 3.4. AMU at Turkey Farms

In addition to the establishment and description of DDD_turkey_ and DCD_turkey_, this study also quantified AMU in 20 French, 20 German and 20 Spanish turkey farms. As far as the authors are aware, this is the first multi-country study reporting on AMU in turkeys at farm level, allowing for comparison of AMU between farms, both within and between countries. Four other studies did look into AMU in turkey production [12,16,17,33]. However, each one only covered one country and two [12,16] used sales data, hampering a more detailed description. One of these studies showed a reduction in AMU at turkey farms in Austria between 2013 and 2019 [17]. Knowing that the data presented here, were collected between 2014 and 2016, it is important to mention that these results will most likely no longer reflect the current situation. However, they do give a valuable benchmark for future studies to identify changes and trends in AMU on turkey farms.

All different dose-based AMU indicators showed high correlations, except for TI based on used daily dose (UDD) and DCD, both at treatment and flock level (Appendix A), which are similar findings as discussed in a previous study [34]. This shows that DDD_vet_ for broilers could be used as a proxy for DDD_turkey_ to quantify AMU on turkey farms. However, noticeable differences were present between the results from TIDDD_vet_-FL and TIDDD_turkey_-FL_._ For example, TIDDD_vet_-FL of the highest using farm from country H was 14% lower than its TIDDD_turkey_-FL_._ This can be explained by the amoxicillin (*n* = 4) and colistin (*n* = 4) treatments on that farm, that have a DDD_turkey_ that is 11.9% and 33.3% lower than its corresponding DDD_vet_, respectively (Figure 1). On average, DDD_turkey_ was 8.8% lower than the corresponding DDD_vet_. As a result, TI at flock level was lower when based on DDD_vet_ compared to TIDDD_turkey_. This shows that TIDDD_vet_ quantification could underestimate AMU on turkey farms with a specific AMU profile.

TI results show big variation in AMU between the flocks, both within and between countries. All countries had at least one flock that did not receive any antimicrobial treatments, while other flocks stood out due to higher usage. This shows potential for reducing AMU on turkey farms. Both an Italian (2015–2017) and Austrian (2013–2019) study already reported on the reduction of AMU in turkeys [16,17]. When comparing the results with a similar study on 181 broiler farms in 9 European countries (median TIDDD_vet_ (95% CI) = 9.0 (5.5–10.8)) [34], AMU in turkeys seems to be slightly higher (median TIDDD_turkey_ (95% CI) = 10.0 (8.0–11.9)). On one hand, other studies confirm this and report a four times higher [16], 2.4 times higher [35], 2 times higher [36] and 1.7 times higher [15] AMU in turkeys. On the other hand, a Canadian [12] and an Austrian [17] study both report an AMU in turkeys that is, four and 0.6 times lower, respectively, compared to broilers. However, these differences are difficult to interpret due to dissimilarities in used methodologies. Knowing that turkeys have a higher predisposition for disease and a longer production cycle, leading to build-up of litter or more environmental stressors, a higher AMU in turkeys compared to broilers is expected [16,37,38,39].

Three antimicrobial classes represented 72.2% of all registered AMU in this study. Similar results were discussed by ESVAC [8] and in studies on pigs [40] and broilers [34]. In fact, the latter also reported ES penicillins, polymyxins and fluoroquinolones as the three biggest classes on broiler farms. Both an Italian and Austrian study in turkeys also showed high usage of penicillins (not split up in ES penicillins and βLS penicillins) and polymyxins [16,17]. However, country differences are present. For example, βLS penicillins were only used in country B. Furthermore, fluoroquinolone use was much lower in country E, which was also the case in an Austrian [17] and Canadian [12] study. In the latter, this was probably driven by the recent implementation of an AMU strategy in Canada, focusing on the elimination of fluoroquinolones in poultry. Instead, they reported their highest use in streptogramins and bacitracin. Both antimicrobials were added to the feed and were not reported in this study. All this suggests that country differences are driven by the available antimicrobial portfolio in a country, combined with national initiatives from stakeholders or governments to reduce AMU, especially when it comes to those antimicrobials that are critical for human medicine. The latter were well represented in this study, with 82.9% of the total AMU, and should be the focus when aiming for a reduction in AMU on turkey farms.

Treatments administered in the first nine weeks represent the majority (79.4%) of all AMU registered in this study. Previous studies in other intensive livestock production systems, such as pigs [40] and broilers [34], also found the focus of AMU in the beginning of production. Compared to broilers, the percentage of flocks getting a treatment in the first week is lower. This might be due to the fact that newly hatched poults could be set-up in brooder rings for the first week [41]. Knowing that disease outbreaks in the first weeks are, among others, related to vertically transmitted infections and bad sanitation in the hatchery [42], a further spread of disease might be hampered thanks to this set-up in smaller groups, possibly leading to a lower AMU. Subsequently, AMU increases in week two, going from 18.3% to 30.0% of treating farms to increase again in week five to 41.7%. This strongly suggests that treatments tend to be initiated at strategic time points, as the poults are taken out of the brooder ring set-up after around 5 to 6 days. They then finish the brooder phase around 5 weeks of age, depending on the sex of the bird, to switch to the growing phase [41]. Unfortunately, we did not collect the necessary information from the farmers to confirm these assumptions. However, a previous study in pigs did also report peaks in AMU in transition periods [40]. Finally, in the growing phase, AMU decreased but remained at a high level to drop below 10% of treating farms in week 15. In this period, respiratory disorders and colibacillosis both peaked, but also intestinal disorders remained a reason to set-up treatment (Appendix A). Most likely this usage can be related to the genetic selection for fast growing animals which has made the modern breeds of turkeys less resilient against infections [42]. Yet again, caution is advised when making these assumptions as other factors might also play their part.

## 4. Materials and Methods

All data were collected within the EU-FP7 EFFORT project. This project studied the epidemiology and ecology of AMR in food-producing animals, the environment and humans, to quantify AMR exposure pathways for humans.

### 4.1. Assigning Defined Daily Doses Turkey (DDD_turkey_) and Defined Course Dose (DCD_turkey_) for France, Germany and Spain

#### 4.1.1. Data Collection

In July 2017, all antimicrobial products registered for use in turkeys in France, Germany and Spain were listed. This information was obtained from official online databases managed by the national regulatory institutions involved in the authorization and registration of veterinary medicinal products. In France this online database is the Index des Médicaments Vétérinaires Autorisés en France (IRCP) [30]; in Germany it is called Veteriärmedizinischer Informationsdienst für Arzneimittelanwendung, Toxikologie und Arzneimittelrecht (Vetidata) [29]; and in Spain this information is accessible online via the Centro de Información online de Medicamentos Veterinarios (CIMAvet) [43]. Only products with an official authorization for use in turkeys were taken into account. Meaning that the following words had to be mentioned as a target species in the national databases: “dinde” or “dindon” for France, “puten” for Germany and “Pavos de engorde” or “pavos reproductores” for Spain. Products registered for use in poultry, chicken or broilers without an additional registration for turkeys were not taken into account. For each registered product, detailed information was collected on the active substance, the administration route, recommended dose and treatment duration. In total, three administration routes were defined: feed, water and parenteral. Within this study, antimicrobials are defined as products with an antimicrobial mechanism and having QJ as first level code in the Anatomical Therapeutic Chemical classification system for veterinary medicinal products (ATCVet system) from the WHO [44]. Products containing zinc oxide or anticoccidial products were not included in this study.

A DDD, expressed in mg/kg body weight per day, was calculated for each authorized product based on the mean recommended dose from the country-specific SPC. When a recommended dose was given as a range, the lower value was seen as the minimal recommended dose and the higher value was seen as the maximal recommended dose. Based on these two values, a mean recommended dose was calculated. When an AS was expressed in international units (IU) in the SPC, the recommended dose was recalculated to mg/kg based on the conversion factors published by ESVAC [45]. When the SPC of a product contained both a recommended dose for prevention in an infected environment and a recommended dose for treatment, only the latter was taken into account. In addition to the recommended dose, treatment duration was also listed, allowing for the calculation of DCD, expressed in mg/kg body weight per treatment course. Again, when the SPC provided a range for the treatment duration, the mean of the minimal and maximal number of treatment days was calculated. The mean treatment duration and the mean recommended dose were then multiplied to become a DCD value. In case of combination products, the DDD value was calculated as the sum of the DDD for each separate AS. Long-acting factors were not assigned to any of the products, as no long-acting products were registered for use in turkeys. Finally, the critical importance of a product was registered based on the guidelines from both the WHO [46] and the OIE [47].

#### 4.1.2. Calculation of Defined Daily Dose Turkey (DDD_turkey_) and Defined Course Dose (DCD_turkey_)

The methodology used for the calculation of DDD_turkey_ and DCD_turkey_ values is derived from a study by Postma and colleagues [22] and the ESVAC report on the principles on assignment of defined daily dose for animals and defined course dose for animals [21]. Each product was categorized based on its combination of ASs and administration route. All products with the same unique combination of ASs and administration route were grouped to calculate the mean, median and mode for the DDD value of that category. The mean value equaled the consensus DDD or DDD_turkey_. This was also done at country level to result in DDD_turkey-National_, in this case each country has its own DDD_turkey_ values for each unique combination of ASs and administration route. Similar to the DDD_turkey_ calculations, DCD values were used to become DCD_turkey_ for each unique combination of ASs and administration route. Finally, differences between the mean recommended dose of the individual products and their corresponding DDD_turkey_ value were evaluated. In addition, differences were also observed between, on one hand the DDD_turkey_ and DCD_turkey_ values, and on the other hand their corresponding DDD_vet_ and DCD_vet_ values for broilers, as calculated by ESVAC [18]. DDD_turkey_ and DCD_turkey_ values were only calculated for active substances that were used on the 60 participating farms in this study.

### 4.2. Quantification of Antimicrobial Usage at Flock Level on 60 Turkey Farms in France, Germany and Spain

#### 4.2.1. Study Sample and Data Collection

In a cross-sectional study, AMU data was collected at 20 turkey farms in each of the participating countries: France, Germany and Spain between October 2014 and October 2016. The selection of farms was subject to inclusion criteria to obtain a homogenous group of farms, both within as between countries. All farms were conventional farms with an all-in-all-out system and had to have at least 3000 birds. Farms had to be epidemiologically unrelated to one another, meaning they could not have contact through trade and each farm had only one owner. Farms were selected in agreement with local farming organizations. Wherever possible, regional stratification was implemented. However, this only succeeded in Germany, while farms in France and Spain were concentrated in the region of Brittany and Andalusia, respectively, the main turkey production sites of these countries. As the 20 farms in each country cannot be considered representative for the turkey production in a country, countries were anonymized (letters B, E and H). On each farm researchers filled in a questionnaire with the farmers to collect technical and AMU data from the farm. To reduce interview bias, researchers followed a questionnaire protocol and received a training. Additionally, information regarding each antimicrobial treatment given to the sampled flock (1 house) was registered. These data are referred to as treatment data and covered all antimicrobial group treatments given to the flock from set-up to slaughter. The indication for each treatment was also registered and was based on the expertise of the veterinarian or farmer. Diagnostics could have been used but were not mandatory. More information and relevant parts of the questionnaire can be found in the Appendix A, part B.

#### 4.2.2. AMU Quantification

AMU was quantified following the same methodology as described by Joosten et al. [34]. The latter reports on a similar study in broilers that was part of the same project, the EFFORT project. Similar to that research, treatment incidence was used as indicator to quantify AMU at flock level on turkey farms. The numerator is equal to the total amount of AS administered. In the denominator, five different parameters were used (DDD_turkey_, DCD_turkey_, DDD_vet_, DCD_vet_ and UDD) resulting in five different formulas (TIDDD_turkey_, TIDCD_turkey_, TIDDD_vet_, TIDCD_vet_ and TIUDD). However, the basic structure of each formula remains the same, as shown in Table 4 The meaning behind the different formulas varies slightly. Firstly, TIDDD_turkey/_DDD_vet_ expresses the number of DDD_turkey_ or DDD_vet_ that were administered per 100 animal-days at risk. In other words, TI equals the number of days per 100 animal-days that the flock has received a standard dose of antimicrobials. Simplified, it reflects the percentage of time that a turkey has been treated with antimicrobials in its life. Secondly, TIDCD_turkey/_DCD_vet_ reflects the number of treatment courses, based on DCD_turkey_ or on DCD_vet,_ per 100 animal-days at risk. Finally, the formula for TIUDD is a simplified version of TIDDD_turkey_ as here UDD appears both in the nominator and denominator. However, the meaning is the same, only here it represents the number of UDD instead of DDD that were administered per 100 animal-days at risk. To calculate a TI at flock level, TIs of all treatments from one flock were added up to obtain TIDDD_turkey_-FL, TIDCD_turkey_-FL, TIDDD_vet_-FL, TIDCD_vet_-FL.

The values for DDD_turkey_, DCD_turkey_, DDD_vet_ and DCD_vet_ are shown in Table 1. DDD_vet_ and DCD_vet_ values for broilers are defined by ESVAC and can be used as a proxy for other poultry species [18]. Calculations for DDD_turkey_ and DCD_turkey_ have been described above (see Section 4.1). ‘Kg of animal at risk’ is the product of ‘standard weight’ and ‘number of animals at the start of the sampled flock’. The latter was retrieved from the questionnaire. The standard weight was set to 6 kg [11]. ‘Number of days at risk’ equaled the duration of the rearing period at each farm. Due to the fact that antimicrobials were given in the feed or via water, UDD was sometimes expressed per ‘kg of feed’ or ‘liter of water’. For this reason, assumptions were made for a standard daily feed intake (300 g) and a standard daily water intake (540 mL) for a turkey with a standard weight of 6kg [48].

### 4.3. Data Processing and Data Analysis

Processing of the AMU data was done following the protocol from the EFFORT project as described in previous publications [34,40]. Data were entered into EpiData version 3.1 software (EpiData Association, Odense, Denmark) by the researchers who visited the farms. Data quality checks were performed using ActivePerl 5.24.1 (ActiveState Software Inc., Vancouver, BC, Canada) and SAS 9.4 (SAS Institute Inc., Cary, NC, USA). Subsequently, SAS was used for database management. A more in-depth quality control and descriptive statistics were both done in Excel^®^ 2016. Even after log_10_ transformation, TIs deviated considerably from normality. Therefore minimum, median and maximum are reported, including 95% confidence intervals based on the adjusted bootstrap percentile method with 1000 replicates. Associations between different quantification methods for AMU were estimated by using a Spearman’s rank correlation test, while differences between indicators were identified by a pairwise Wilcoxon rank test in R version 3.4.0 software (https://cran.r-project.org accessed on 7 october 2020).

## 5. Conclusions

This study presents a list of DDD_turkey_ and DCD_turkey_ to allow for a species-specific, on farm quantification of AMU on European turkey farms. Assigning DDD_turkey_ has shown the need for a unified European antimicrobial drug portfolio and harmonization of recommended doses across the SPCs of similar antimicrobial products authorized for use in turkeys, at the least when treating identical diseases. The overview provided in this study can help guide prioritisation and selection of candidates. Additionally, a comparison with DDD_vet_ values from ESVAC, shows that not all DDD_vet_ values for broilers can be applied to turkeys. Therefore, preferably DDD_turkey_ is used when quantifying AMU on European turkey farms. Furthermore, AMU on turkey farms showed great variation regarding the amount used. The majority of the antimicrobials were used in the beginning of production and very often CIAs were used. This shows potential for reducing and improving AMU on turkey farms.

## Figures and Tables

**Figure 1 antibiotics-10-00971-f001:**
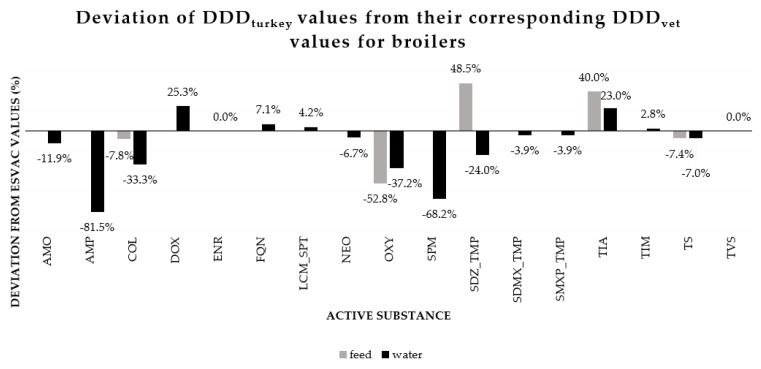
Deviation of DDD_turkey_ values from their corresponding DDD_vet_ values, stratified by unique combinations of active substance and administration route. DDD_vet_ values are set to 0%. A plus sign before the percentage means that the DDD_turkey_ value was higher than its corresponding DDD_vet_ value. A minus sign means that the DDD_turkey_ value was lower. All 0% deviations belong to active substances administered via water. AMO = amoxicillin; AMP = ampicillin; COL = colistin; DOX = doxycycline; ENR = enrofloxacin; FQN = flumequin; LCM_SPT = lincomycin in combination with spectinomycin; NEO = neomycin; OXY = oxytetracycline; SPM = spyramycin; SDZ_TMP = sulfadiazine in combination with trimethoprim; SDMX_TMP = sulfadimethoxine in combination with trimethoprim; SMXP_TMP = sulfamethoxypyridazine in combination with trimethoprim; TIA = tiamuline; TIM = tilmicosin; TS = tylosin; TVS = tylvalosin.

**Figure 2 antibiotics-10-00971-f002:**
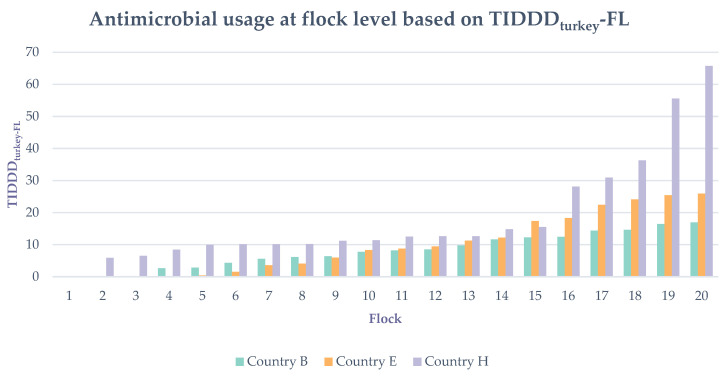
Antimicrobial usage of each of the 20 turkey flocks/country expressed as treatment incidence (TI) at flock level (FL): shown for TI based on DDD_turkey_ and summed up at flock level (TIDDD_turkey_-FL).

**Figure 3 antibiotics-10-00971-f003:**
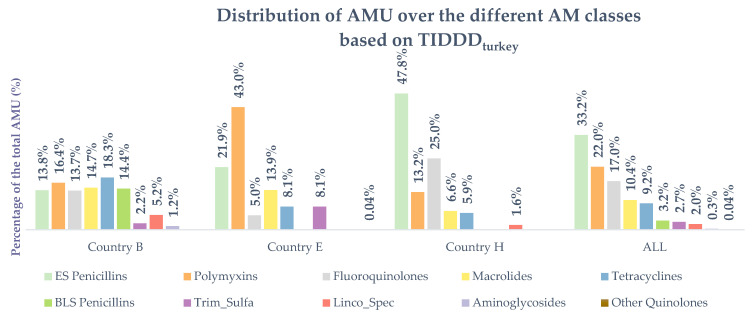
Distribution of antimicrobial usage (AMU) over the different antimicrobial classes, stratified per country and in total AMU is represented by treatment incidence (TI) based on DDD_turkey_ (TIDDD_turkey_). Linco_Spec = lincomycin in combination with spectinomycin. Trim_Sulfa = trimethoprim in combination with sulfonamides.

**Figure 4 antibiotics-10-00971-f004:**
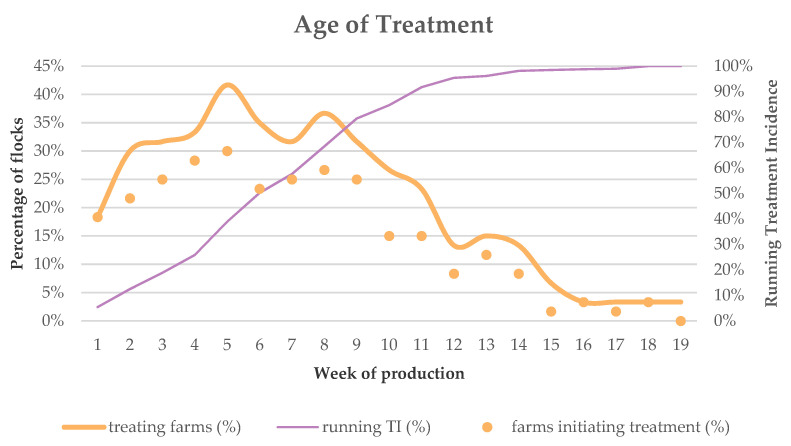
The age of treatment with antimicrobials of the 60 turkey flocks. The dots show the proportion (%) of turkey flocks (on the *y*-axis) where treatments with antimicrobials were initiated in a certain week of production (*x*-axis). The orange full line shows the proportion (%) of turkey flocks (left *y*-axis) that were being treated with antimicrobials in a certain week of production (*x*-axis). The purple full line shows the running treatment incidence (left *y*-axis) over the production cycle in weeks (*x*-axis).

**Table 1 antibiotics-10-00971-t001:** Mean DDD_turkey_ and DCD_turkey_ values (consensus DDD_turkey_ and consensus DCD_turkey_) for each category with a unique combination of active substance and administration route, combined with the corresponding DDD_vet_ and DCD_vet_ values for broilers from the European Surveillance of Veterinary Antimicrobial Consumption (ESVAC), when available [18].

Active Substance	Consensus DDD_turkey_ (*n*)			Consensus DCD_turkey_			DDD_vet_	DCD_vet_
	Feed	Injection	Water	Feed	Injection	Water	Feed/Water	Feed/Water
Amoxicillin	-	-	14.1 (25)	-	-	60.0	16.0	74.0
Ampicillin *	-	10.5 (1)	20.0 (4)	-	31.5	60.0	108.0	453.0
Benzylpenicillin *	-	-	28.8 (1)	-	-	115	-	-
Colistin	4.7 (3)	2.5 (2)	3.4 (31)	32.7	7.5	13.8	5.1	27.0
Doxycycline	-	-	18.8 (37)	-	-	90.4	15.0	61.0
Enrofloxacin	-	-	10.0 (29)	-	-	40.0	10.0	41.0
Flumequine	-	-	15.0 (10)	-	-	66.0	14.0	60.0
Linco_spec	-	30.0 (2)	62.5 (2)	-	90.0	287.2	60.0	112.0
Neomycin	-	-	22.4 (9)	-	-	93.7	24.0	114.0
Oxytetracycline *	18.4 (9)	7.5 (1)	24.5 (18)	180.9	30.0	106.7	39.0	207.0
Phenoxymethylpenicillin **	-	-	16.8 (6)	-	-	83.8	17.0	84.0
Spiramycin	-	-	23.3 (2)	-	-	89.1	73.0	459.0
Sulfadiazine_TMP	60.0 (3)	-	30.7 (5)	420.0	-	165.6	40.4	219.0
Sulfadimethoxine_TMP	-	-	36.0 (7)	-	-	190.3	37.4	161.0
Sulfamethoxypyridazine_TMP *	-	-	28.2 (1)	-	-	113.0	29.4	93.0
Tiamulin	32.2 (4)	-	28.3 (12)	224.8	-	130.2	23.0	88.0
Tilmicosin	-	-	18.5 (6)	-	-	55.5	18.0	53.0
Tylosin	75 (6)	-	75.3 (17)	262.5	-	280.2	81.0	342.0
Tylvalosin	-	-	25.0 (3)	-	-	125.0	25.0	75.0

* (one of the) value(s) is based on the recommended dose from the summary of product characteristics of 1 product. ** no products registered for turkeys, so value is based on all similar products registered for broilers in France, Germany and Spain. (*n*) = the number of products registered within each category. Linco_spec = lincomycin in combination with spectinomycin. _TMP = in combination with trimethoprim. DDD and DCD values are given in mg/kg bodyweight/day.

**Table 2 antibiotics-10-00971-t002:** The top 20 deviating products from the mean DDD_turkey_ value (consensus DDD_turkey_).

No.	Active Substance	Administration Route	Country	Product Name	Consensus DDD_turkey_	Mean Recommended Dose from SPC **	Deviation from DDD_turkey_ (%) *
1	oxytetracycline	water	Germany	Ursocyclin-Pulver^®^ 20%	24.5	80	+227%
2	oxytetracycline	feed	France	VO 31-2 Oxytetracycline^®^	18.4	40	+117%
3	oxytetracycline	feed	France	Oxytetracycline^®^ 40-CR^®^	18.4	40	+117%
4	oxytetracycline	feed	France	PM 17-A Oxytetracycline^®^ 40	18.4	40	+117%
5	oxytetracycline	water	Spain	Terramicina^®^ 55	24.5	50	+104%
6	sulfadimethoxine	water	France	Trimethosulfa^®^ V	36.0	46.7	+58%
7	neomycine	water	Spain	Nemicina^®^	22.4	33.3	+48%
8	colistin	water	Germany	Belacol^®^ 100%	3.4	4.8	+41%
9	colistin	water	Germany	Belacol^®^ 24%	3.4	4.8	+41%
10	colistin	water	Germany	Colistin^®^ 2,4	3.4	4.8	+41%
11	doxycycline	water	France	Ronaxan^®^ P.S. 5%	18.8	10	−47%
12	doxycycline	water	Germany	Doxipulvis^®^ 500	18.8	10	−47%
13	doxycycline	water	Spain	Doxi^®^ 100	18.8	10	−47%
14	doxycycline	water	Spain	Doxipulvis^®^ 500	18.8	10	−47%
15	oxytetracycline	water	Spain	Oxitetraciclina^®^ lagro	24.5	12.5	−49%
16	neomycine	water	Spain	Nisocline^®^	22.4	7.5	−67%
17	neomycine	water	Spain	Neocil^®^ premix	22.4	7.5	−67%
18	oxytetracycline	feed	Spain	Oxitetraciclina^®^ BMP	18.4	10.2	−72%
19	oxytetracycline	feed	Spain	Oxitetraciclina^®^ 100	18.4	10.2	−72%
20	oxytetracycline	feed	Spain	Z-19^®^	18.4	10.2	−72%

* a plus or a minus sign before the percentage means that the recommended dose from this product was respectively higher or lower in comparison with the mean consensus DDD_turkey_. DDD_turkey_ and recommended doses values are given in mg/kg bodyweight/day. ** Summary of product characteristics.

**Table 3 antibiotics-10-00971-t003:** Antimicrobial usage quantified using treatment incidence (TI). TI is summed up at flock level and expressed by TIDDD_turkey-FL_, TIDCD_turkey_-FL, TIDDD_vet_-FL, TIDCD_vet_-FL and TIUDD-FL.

Country (*n*)	NNUF*	TIDDD_turkey_-FL Med [95% CI] (Min-Max)	TIDDD_vet_-FL Med [95% CI] (Min-Max)	TIDCD_turkey_-FL Med [95% CI] (Min-Max)	TIDCD_vet_-FL Med [95% CI] (Min-Max)	TIUDD-FL Med [95% CI] (Min-Max)
B	3	8.0 [4.3–11.6]	6.0 [3.9–11.2]	1.9 [1.1–2.6]	1.3 [0.9–2.9]	2.7 [2.1–3.1]
(20)		(0.0; 16.9)	(0.0; 17.6)	(0.0; 4.0)	(0.0; 4.1)	(0.0; 4.3)
E	3	8.5 [1.5–12.2]	6.0 [1.3–10.5]	2.1 [0.8–3.7]	1.2 [0.3–2.0]	2.4 [1.8–3.1]
(20)		(0.0; 25.9)	(0.0; 17.0)	(0.0; 6.8)	(0.0; 3.9)	(0.0; 5.7)
H	1	12.0 [10.1–15.1]	11.3 [9.0–13.7]	2.9 [2.4–3.5]	2.5 [2.1–3.2]	4.6 [3.5–4.8]
(20)		(0.0; 65.7)	(0.0; 57.8)	(0.0; 16.1)	(0.0; 13.4)	(0.0; 6.6)
Total	7	10.0 [8.0–11.9]	8.7 [6.1–11.4]	2.4. [1.9–2.8]	1.9 [1.4–2.5]	3.0 [2.7–3.6]
(60)		(0.0; 65.7)	(0.0; 57.8)	(0.0; 16.1)	(0.0; 13.4)	(0.0; 6.6)

NNUF* = Number of non-using flocks in treatment dataset, TIDDD/DCD/UDD_turkey/vet_-FL = treatment incidence based on DDD_turkey_/DCD_turkey_/DDD_vet_/DCD_vet_/UDD and summed up at flock level (-FL) DDDvet and DCDvet are values defined for broilers by the European Medicines Agency. Med = median, [95% CI] = 95% CI based on the adjusted bootstrap percentile method with 1000 replicates, Min = mininum, Max = maximum, AM = antimicrobials.

**Table 4 antibiotics-10-00971-t004:** Formulas to quantify antimicrobial usage using treatment incidence (TI).

Formula		Result
TI_DDDturkey_ ^1^	$=\frac{\mathrm{UDD}\;(\mathrm{mg}/\mathrm{kg}/\mathrm{day})\;\times \;\mathrm{treatment}\;\mathrm{duration}\;\left(\mathrm{days}\right)\;\times \;\mathrm{nr}.\;\mathrm{of}\;\mathrm{animals}\;\mathrm{treated}\;\times \;\mathrm{standard}\;\mathrm{weight}\;\mathrm{per}\;\mathrm{animal}\;\left(\mathrm{kg}\right)}{\mathrm{DDDturkey}\;(\mathrm{mg}/\mathrm{kg}/\mathrm{day})\;\times \;\mathrm{nr}.\;\mathrm{of}\;\mathrm{days}\;\mathrm{at}\;\mathrm{risk}\;\left(\mathrm{days}\right)\times \;\mathrm{kg}\;\mathrm{of}\;\mathrm{AAR}\;\left(\mathrm{kg}\right)}\times 100\;\mathrm{AAR}$	Nr. of DDD_turkey_/100 days at risk/day
TI_DCDturkey_ ^2^	$=\frac{\mathrm{UDD}\;(\mathrm{mg}/\mathrm{kg}/\mathrm{day})\;\times \;\mathrm{treatment}\;\mathrm{duration}\;\left(\mathrm{days}\right)\;\times \;\mathrm{nr}.\;\mathrm{of}\;\mathrm{animals}\;\mathrm{treated}\;\times \;\mathrm{standard}\;\mathrm{weight}\;\mathrm{per}\;\mathrm{animal}\;\left(\mathrm{kg}\right)}{\mathrm{DCDturkey}\;(\mathrm{mg}/\mathrm{kg}/\mathrm{day})\;\times \;\mathrm{nr}.\;\mathrm{of}\;\mathrm{days}\;\mathrm{at}\;\mathrm{risk}\;\left(\mathrm{days}\right)\;\times \;\mathrm{kg}\;\mathrm{of}\;\mathrm{AAR}\;\left(\mathrm{kg}\right)}\times 100\;\mathrm{AAR}$	Nr. of DCD_turkey_/100 days at risk
TI_UDD_	$\frac{\;\mathrm{treatment}\;\mathrm{duration}\;\left(\mathrm{days}\right)\;\times \;\mathrm{nr}.\;\mathrm{of}\;\mathrm{animals}\;\mathrm{treated}}{\;\mathrm{nr}.\;\mathrm{of}\;\mathrm{days}\;\mathrm{at}\;\mathrm{risk}\;\left(\mathrm{days}\right)\;\times \;\mathrm{no}.\;\mathrm{of}\;\mathrm{AAR}}\times 100\;\mathrm{AAR}$	Nr. of UDD/100 days at risk/day

UDD = used daily dose; Nr = number; DDD_turkey_ = defined daily dose for turkeys; DCD_turkey_ = defined course dose for turkeys; AAR = animals at risk; ^1^ When you switch DDD_turkey_ by DDD_vet_ within the TIDDD_turkey_ formula, you have the formula for TIDDD_vet_; ^2^ When you switch DCD_turkey_ by DCD_vet_ within the TIDCD_turkey_ formula, you have the formula for TIDCD_vet_. DDD_vet_ and DCD_vet_ are values defined for broilers by the European Medicines Agency.

## Data Availability

The data presented in this study are available upon request from thecorresponding author.

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
