# Peer review of "Assigning Defined Daily/Course Doses for Antimicrobials in Turkeys to Enable a Cross-Country Quantification and Comparison of Antimicrobial Use"

_antibiotics, 2021, doi:10.3390/antibiotics10080971_

Round 1
Reviewer 1 Report
The article compares the antimicrobial substance usage in turkeys flocks in three European countries. It is wll writtn work, but I hav som doubts.
Why the products registered for poulty other than turkeys were not calculated? The substances were used, so the antimicrobial resistance in bateria could devloped. Were among not mentioned antimicrobials any critically important for human medicine?
You showed the diseases which were treated in turkeys - were the bacteria isolatd and antibiotic sensitivity tests done?
Author Response
Comments Reviewer 1:
The article compares the antimicrobial substance usage in turkeys flocks in three European countries. It is wll writtn work, but I hav som doubts.
Why the products registered for poulty other than turkeys were not calculated? The substances were used, so the antimicrobial resistance in bateria could devloped.
- Thank you for this comment. It is indeed correct that for some active substances there are no registered products for turkeys and products with a registration for another poultry species can be used. However, there was only one treatment with phenoxymethylpenicillin, an active substance category without any products registered for use in turkeys (in one of the three countries). In this case, we used the SPC’s from broiler products to calculate a DDD for Spain, France and Germany, to at least be consistent regarding that part of the methodology (see Table 1, subscript). To avoid misinterpretation on this issue, we added a more detailed description on the relative importance of phenoxymethylpenicllin treatments (n=1) within the group of the 23 treatments with β-lactamase sensitive penicillins (see line 246-247). Furthermore, it was our aim to develop turkey-specific DDD-values to enable a more precise AMU quantification. For this reason, we only looked at products that are registered for use in this species. It is possible that some products that are used on turkey farms are only registered for other poultry species. However, as mentioned earlier, our data show that this is rather limited. For all used active substances, except one (phenoxymethylpenicillin), did we find products registered for use in turkeys. knowing that pharmacokinetics can differ between turkeys and broilers, we believe that DDD-values should be based on products registered for that species and that species alone, instead of including products authorized for use in other poultry species, that can have a totally different recommended dose (see example f ampicillin lines 395-403)
Were among not mentioned antimicrobials any critically important for human medicine?
- Thank you for this comment. If antimicrobials were not mentioned, such as 3rd and 4th generation cephalosporins, these products were not used on the 60 farms in this study. Therefore, we did not report on these, in neither parts of the paper (calculation DDD-values and AMU quantification results). To make this clear to the reader, we added the following sentence to the main text: “DDDturkey and DCDturkey values were only calculated for active substances that were used on the 60 participating farms in this study.” ‘see line 556-557)
You showed the diseases which were treated in turkeys - were the bacteria isolatd and antibiotic - sensitivity tests done?
- We agree with the Reviewer that such data would have been of interest. Ideally diagnosis would have been made based on isolation in combination with an antibiotic sensitivity test. However, in the field, antibiotics are often given based on the clinical symptoms without any additional diagnostics. To represent this situation, we did not make it mandatory to do a diagnostic or sensitivity test, as this might influence the AMU results of the flock. Most likely, indication of treatment was based on the expertise of the veterinarian (or farmer). We do not know if/when diagnostics were performed, as this information was not collected. To make this clear to the reader we added the following sentence to the M&M: “The indication for each treatment was also registered and was based on the expertise of the veterinarian or farmer. Diagnostics could have been used but were not mandatory.” (lines 578-580)
- We did isolate E.coli, an indicator bacteria for AMR, from faecal samples, collected as closely to slaughter as possible. Afterwards, these isolated bacteria were tested on AMR, using multiple methods (disk diffusion, PCR, WGS). Results on this aspect of our research can be found in the following paper: https://www.mdpi.com/2079-6382/10/7/820
Reviewer 2 Report
The content of the manuscript is highly significant for the future of antibiotic use reporting in general and particularly for generation of adequate data for turkey. The importance of species-specific DDD values highlighted by the authors is crucial given the major differences of turkey meat production systems compared to that of broilers. The collection of data, calculations are appropriate, and results are presented in detail. My comments on this review are suggestions for minor changes.
The main drawback in the design is the non-randomization/representability of the farms included in the study. The authors accordingly present farm production parameters to overcome this issue, but interpretation is difficult without a reference of the average parameters in European turkey farms, or at least in the countries included in the study. Other farm information from the local farming organizations that helped recruit the farms, and/or author contribution about how the final selection was made would be helpful to understand the reported AMU.
A lot of emphasis is given to the deviations from DDDvet and DDDturkey values. As pointed out by the authors, the only studies reporting AMU in turkey seem to have calculated their own country based DDD values. Therefore, deviations between DDDvet and DDDturkey highlight the need for the work presented here, but the importance of the numerical differences per se are not critical. On the other hand, it would be interesting to know how differing your DDDturkey values are from those turkey-specific reported values in previous studies in other European countries.
In general, the manuscript is rather extent and for readability I would reduce/simplify text.
Other minor points:
1. Note that some acronyms are presented for the first time in material and methods, but they are used in discussion (which comes before in the structure of the paper). I noticed UDD, but there may be others.
2. Homogenize subscript of vet/turkey after DDD.
3. It is a bit hard to see which AS corresponds to each bar in figure 1. Perhaps x-axis labels at 90 degrees instead of 45 and removing empty feed columns would help? Also, the legend should probably mention that the 0% columns are data for water.
4. Line 133, it may be easier to just mention the two AS without DDDvet excluded from figure 1.
5. Figure 2. One flock was included per farm, correct? So “flock” and “farm” refer to the same?
6. Line 150 is interrupted.
7. DDDA in table S2 stands for?
Author Response
Comments Reviewer 2
The content of the manuscript is highly significant for the future of antibiotic use reporting in general and particularly for generation of adequate data for turkey. The importance of species-specific DDD values highlighted by the authors is crucial given the major differences of turkey meat production systems compared to that of broilers. The collection of data, calculations are appropriate, and results are presented in detail. My comments on this review are suggestions for minor changes.
- Thank you for this positive feedback and taking the time to read through the manuscript in detail to address points for improvement!
The main drawback in the design is the non-randomization/representability of the farms included in the study. The authors accordingly present farm production parameters to overcome this issue, but interpretation is difficult without a reference of the average parameters in European turkey farms, or at least in the countries included in the study. Other farm information from the local farming organizations that helped recruit the farms, and/or author contribution about how the final selection was made would be helpful to understand the reported AMU.
- Thank you for this comment. The authors agree with the reviewer that the non-random sample is a weakness of the study. As this is important for the representability of the results, we included as much details on the selection of the farms as possible and also emphasize this aspect in lines 190-195. A reference would indeed be helpful to interpret the farm production parameters. For this reason we included Table S3-bis in the supplementary materials, which is based on data from Eurostat (data-explorer). This table shows data on the size of poultry farms in the three countries, other than broilers and laying-hens. This includes turkeys but also duck, guinea fowl and other minor poultry species. Specific numbers for turkey farms are not available from Eurostat. Nor did we find suitable data in national reports (such as the following for France: https://www.volaille-francaise.fr/wp-content/uploads/rapport2019chiffrescles.pdf). We believe this table can help as a reference, although ideally this data would be more recent (now 2013) and only focused on turkey production.
A lot of emphasis is given to the deviations from DDDvet and DDDturkey values. As pointed out by the authors, the only studies reporting AMU in turkey seem to have calculated their own country based DDD values. Therefore, deviations between DDDvet and DDDturkey highlight the need for the work presented here, but the importance of the numerical differences per se are not critical. On the other hand, it would be interesting to know how differing your DDDturkey values are from those turkey-specific reported values in previous studies in other European countries.
- Thank you for this comment. We agree with the reviewer that it would be interesting to compare DDDturkey values with the DDD values from other turkey studies. However, these papers do not always report the DDD values, but only show the AMU results. Studies that do report the DDD values make a comparison challenging, as for example in the Italian study (Caucci et al. 2019), they report only one list of DDD values for both broilers and turkeys. The Canadian studies (Agunos et al 2017, 2019) report on totally different categories of unique combination of active substance and administration route, as AMU on Canadian turkey farms seems very different from European turkey farms (discussed in lines 460- 469). In addition, an in depth comparison with available DDD values from other studies, would make this paper even longer, which is something we would want to avoid. This is probably also the reason why other papers do not report in such detail on their DDD values, as this results in a longer paper. However, we believe this is a crucial part of the research and valuable information for future research. We hope our general discussion in 3.2 and 3.3 is sufficient enough. Here we address possible country effects and species effects in general.
In general, the manuscript is rather extent and for readability I would reduce/simplify text.
- We agree with the reviewer that this manuscript is rather long. However, this is a result of combining two papers into one (as mentioned earlier). Other researchers often published the DDD(animal) calculations as a separate paper, followed by a paper that reports on the AMU results where these DDD values were used in the quantification (Agunos et al., Postma et al., Caucci et al.). We could split this paper into two separate ones, but would prefer to keep it as it is. We are convinced that the two parts together give a more nuanced view on the results. For example, the available antimicrobial drug portfolio in a country, which is discussed in the first part (calculation of DDDturkey) is strongly connected with the AMU results from the second part as mentioned in lines 466-469. We hope the reviewer and the editor can support us in this matter. However, we did review the text for unnecessary repetitions and tried to write as concise as possible to keep the length of the text to a minimum.
Other minor points:
1. Note that some acronyms are presented for the first time in material and methods, but they are used in discussion (which comes before in the structure of the paper). I noticed UDD, but there may be others.
- Thank you for noticing this. We made the necessary changes so acronyms are only used after the full presentation of the word. (see track changes throughout the document).
- Homogenize subscript of vet/turkey after DDD.
- Thank you for noticing this. We made the necessary changes so this is consistent throughout the manuscript. (see track changes throughout the document).
- It is a bit hard to see which AS corresponds to each bar in figure 1. Perhaps x-axis labels at 90 degrees instead of 45 and removing empty feed columns would help? Also, the legend should probably mention that the 0% columns are data for water.
- Thank you for these suggestions, the figure was adjusted accordingly and the following sentence was added: “All 0% deviations belong to active substances administered via water.” (line 141)
- Line 133, it may be easier to just mention the two AS without DDDvet excluded from figure 1.
- Thank you for this suggestion, this is indeed easier to understand as a reader. The manuscript now reads: “However, benzylpenicillin and phenoxymethylpenicillin are not included, as the former has no corresponding DDDvet value and the latter did not have products registered for use in turkeys.” (lines134-136)
- Figure 2. One flock was included per farm, correct? So “flock” and “farm” refer to the same?
- Thank you for this comment, it is indeed correct that one flock per farm was included, so flock and farm refer to the same. To avoid misinterpretation, we changed the text underneath the figure to the following: “Figure 2. Antimicrobial usage of each of the 20 turkey flocks/country expressed as treatment incidence (TI) at flock level (FL): shown for TI based on DDDturkey and summed up at flock level (TIDDDturkey-FL).” (see lines 233-234)
- Line 150 is interrupted.
- Thank you for this comment, this part should have been removed. We changed the text acoordingly.
- DDDA in table S2 stands for?
- Thank you for this question, this acronym was not yet explained. DDDA stands for defined daily dose animal and is the same as DDDturkey, We used it here (instead of DDDturkey) to avoid a larger column width that would no longer make the table fit on 1 page. The following was added to the description of the table: “DDDA stands for defined daily dose animal and is the same as DDDturkey.”.